# In Vivo Metabolic Response upon Exposure to Gold Nanorod Core/Silver Shell Nanostructures: Modulation of Inflammation and Upregulation of Dopamine

**DOI:** 10.3390/ijms21020384

**Published:** 2020-01-08

**Authors:** Haiyun Li, Tao Wen, Tao Wang, Yinglu Ji, Yaoyi Shen, Jiaqi Chen, Haiyan Xu, Xiaochun Wu

**Affiliations:** 1CAS Key Laboratory of Standardization and Measurement for Nanotechnology & CAS Center for Excellence in Nanoscience, National Center for Nanoscience and Technology, Beijing 100190, China; lihaiyun@nanoctr.cn (H.L.); jiyl@nanoctr.cn (Y.J.); chenjq@nanoctr.cn (J.C.); 2University of Chinese Academy of Science, Beijing 100049, China; 3Institute of Basic Medical Sciences, Chinese Academy of Medical Sciences & Peking Union Medical College, Beijing 100730, China; went@ibms.pumc.edu.cn (T.W.); 18514236215@163.com (T.W.);

**Keywords:** gold nanorod core/silver shell nanostructures, inflammation, metabolism, dopamine

## Abstract

With the increasing applications of silver nanoparticles (Ag NPs), the concerns of widespread human exposure as well as subsequent health risks have been continuously growing. The acute and chronic toxicities of Ag NPs in cellular tests and animal tests have been widely investigated. Accumulating evidence shows that Ag NPs can induce inflammation, yet the overall mechanism is incomplete. Herein, using gold nanorod core/silver shell nanostructures (Au@Ag NRs) as a model system, we studied the influence on mice liver and lungs from the viewpoint of metabolism. In agreement with previous studies, Au@Ag NRs’ intravenous exposure caused inflammatory reaction, accompanying with metabolic alterations, including energy metabolism, membrane/choline metabolism, redox metabolism, and purine metabolism, the disturbances of which contribute to inflammation. At the same time, dopamine metabolism in liver was also changed. This is the first time to observe the production of dopamine in non-neural tissue after treatment with Ag NPs. As the upregulation of dopamine resists inflammation, it indicates the activation of antioxidant defense systems against oxidative stress induced by Au@Ag NRs. In the end, our findings deepened the understanding of molecular mechanisms of Ag NPs-induced inflammation and provide assistance in the rational design of their biomedical applications.

## 1. Introduction

Silver nanoparticles (Ag NPs) have been commercialized for application in daily products and medical products because of its good antimicrobial activity [1,2]. With the increasing possibility of humans contacting Ag NPs directly or indirectly, the potential influence of Ag NPs on human health is always a hot issue in focus. The small size allows nanoparticles to more easily reach various organs. The previous publication shows that the liver is one of main target organs which Ag NPs would depose in [3].

Most of the toxicity of Ag NPs relies on the generation of reactive oxygen species (ROS), which may disrupt mitochondrial function, destroy cell lysosomes and membranes, and then lead to cell death [4,5,6,7]. The toxicity after acute or chronic exposure to Ag NPs has been investigated. Studies in vivo have demonstrated that Ag NPs can induce inflammation and hepatotoxicity [8,9]. However, the specific mechanism of Ag NPs’ induced inflammation is still incomplete. Metabolism can act as a powerful means to systematically study the mechanism of toxicological and pathological changes [10]. Metabolomic changes can sensitively reflect the toxicological effects of metal oxide NPs [11]. Metabolomic profiling is used to reveal the molecular mechanisms for the adverse effects of zinc oxide NPs and nanopolystyrene [12]. Even without obvious toxicity, nanoparticles may cause metabolic alterations. Given that chronic inflammation is present in a large number of metabolic disorders, interaction between inflammation and metabolism are intricate and complex. Citrate, as an important intermedia metabolite of energy metabolism, plays a significant role in immune response [13]. Lipogenesis regulates inflammation by affecting the phospholipid component of the cell membranes [14]. Choline participating in choline metabolism as well as membrane phospholipids metabolism is reported to modulate the NLRP3-dependent inflammation [15]. In addition, dopamine can restrain inflammation and inhibit the release of pro-inflammatory cytokines [16]. It has been reported that Ag NPs can also cause the changes of metabolic profile containing metabolites associated with energy metabolism, antioxidant defenses, amino acid metabolism, and lipid metabolism [17]. Ag NPs can provide a platform to explore the affiliation between inflammation and metabolism in vivo. As is well known, dopamine plays a vital role in nerve conduction. The level of dopamine is related to tyrosine hydroxylase. It is reported that some nanoparticles can affect the tyrosine hydroxylase in neurocytes. Titanium dioxide nanoparticles result in downregulation of tyrosine hydroxylase [18]. Ceria nanoparticles protect the tyrosine hydroxylase by scavenging ROS [19]. Ag NPs’ exposure can lead to the increased gene expression of tyrosine hydroxylase [20]. Dopamine also exists in non-neural tissue and participates in immunoregulation. The anti-inflammatory effect of dopamine has been reported. In addition, dopamine can affect cellular oxidative-redox homeostasis by promoting iron accumulation in macrophages [21]. However, it is unclear how dopamine responds to nanoparticles in non-neural tissue.

Here, gold nanorods core/silver shell nanostructures (Au@Ag NRs) were selected as a model of Ag NPs based on the following considerations. First, due to the high chemical stability of Au, we could use Au as an internal reference to estimate the release of silver ions based on the change of Ag/Au ratio. Second, compared to spherical NPs, NRs have aspect ratio-dependent local plasmonic features, which can be extended to a near infrared ray (NIR) spectral region and thus are beneficial for biological applications. Third, the unique rod-shape makes it more easily identify in vitro. The method of multiple exposures was used to explore the potential risk of chronic exposure to a low dose of Ag NPs. Metabolism approaches were used to measure the metabolites’ alterations induced by Au@Ag NRs. We found out that Au@Ag NRs resulted in alterations not only in energy metabolism, membrane metabolism, choline metabolism, but also in dopamine metabolism. Moreover, histochemical studies and macrophages-based tests showed that Au@Ag NRs stimulated inflammation. These results exhibited a relationship between metabolism and pathogeny of inflammation and a linkage between Ag NP exposure and immune behavior.

## 2. Results and Discussion

### 2.1. Characterizations of Au@Ag NRs

Herein, Au@Ag NRs with sizes similar to a previous study were used, as they were shown to accumulate in organs such as liver and lung by intravenous injection [22]. The morphology of Au@Ag NRs was characterized using transmission electron microscopy (TEM). From the TEM images (Figure 1A and 1B), the obtained nanorods showed a cuboid shape with a core–shell structure. The inner darker part was an Au NR core and the outer shallow part was an Ag shell. ImageJ (ImageJ 1.48, NIH, Bethesda, MD, USA) was used to obtain the sizes of synthesized NRs (Appendix A). According to the statistical results of rod-shaped particles, the NRs had a mean length of 78.0 ± 4.1 nm (with a relative standard deviation of 5.3%) and a mean width of 32.4 ± 1.7 nm (with a similar relative standard deviation of 5.2%), respectively. The mean aspect ratio was 2.41 ± 0.16 with a relative standard deviation of 6.6%. In all, the obtained NRs exhibited a relatively narrow size distribution. The formation of the Au core–Ag shell nanostructure was further verified by EDS (energy disperse spectroscopy) element mappings of an individual NR as shown in Figure 1C. The weight ratio of silver to gold was 0.71 as quantified by inductively coupled plasma mass spectrometer (ICP-MS). The NRs used for study had a two-layer organic coating of an inner CTAB (cetyl trimethyl ammonium bromide) bilayer and an outer sodium polystyrene sulfonate (PSS) polyelectrolyte layer via electrostatic interactions and therefore exhibited a good dispersion stability in water. The suspension of Au@Ag NRs in glucose solution (5% weight) exhibited a narrow longitudinal surface plasma resonance (LSPR) band peaked at 645 nm as that in water (Figure 1D). Owing to a “plasmonic-focusing” effect [23], the LSPR band of Au@Ag NRs was narrower than that of the inside Au NR cores, which is beneficial for many plasmonic applications, such as sensing and multiplex labeling. 

### 2.2. Effect of Au@Ag NRs on Metabolomics in Mouse Liver

As shown in Table 1, seven days after multi-injections of Au@Ag NRs, we could still detect the silver in liver. In order to study the metabolic alterations in mouse liver after multi-injections of Au@Ag NRs at a low dose, liquid chromatography-mass spectrometry (LC-MS) technology was used to analyze the liver extracts in positive and negative modes, respectively. Based on the data of the mass-to-charge ratio (m/z) signals, an orthogonal partial least squares discrimination analysis (OPLS-DA) was further performed. The model discriminated metabolic profiles into two groups. The results were showed in the score plots (Figure 2A, Appendix A), indicating that the multiple exposures of Au@Ag NRs induced the metabolic variation. Furthermore, variable importance for the projection (VIP) obtained from OPLS-DA was used to access the differential metabolites. The metabolites with VIP values more than 1 were considered as differential metabolite candidates. Furthermore, metabolites with VIP > 1 and *p*-value < 0.05 were metabolites with significant difference while metabolites with VIP > 1 and 0.05 < *p*-value < 0.1 were differential metabolites as shown in Appendix A. Heat maps were used to present the relative abundance of the significantly differential metabolites (Figure 2B, Appendix A). Glucose, lactate, and malic acid are related with energy metabolism, which declined remarkably after multiple Au@Ag NRs exposure. It implied that Au@Ag NRs affected the energy metabolism. 

The levels of phosphatidylcholine, lyso-phosphatidylcholines, glycerophosphocholine, phosphorylcholine, and choline, which derive from membrane metabolism and choline metabolism, were disturbed after treatment with Au@Ag NRs. The observed changes in the above metabolites underlined that Au@Ag NRs resulted in different metabolic disruptions in mouse liver (Table 2, 27 metabolite changes: 17 downregulated and 10 upregulated). Similarly, it is reported that Ag NPs can give rise to the decline in mitochondrial membrane potential, the impairment of lysosomal membrane, and even lactate dehydrogenase leakage [4,5,24,25]. Mitochondrial membrane potential is related with electron transport chain in mitochondrion, further reflecting the state of energy metabolism. The leakage of lysosomal membrane may cause the release of cathepsin B which can activate the NLRP3 inflammasome [26]. 

Interestingly, it is noted that tyrosine and dopamine upregulated after multiple Au@Ag NR treatment. Both of them were associated with the synthesis of dopamine. The results indicated that Au@Ag NRs altered the synthesis of dopamine. Dopamine is not only a neurotransmitter, but also an immune regulator. Dopamine has been reported to possess anti-inflammatory function, suppressing the activation of NLRP3 inflammasome and influencing the generation of pro-inflammatory cytokines [16,27]. In addition, cholic acid was also increased significantly, which could inhibit the activation of NLRP3 inflammation [15].

### 2.3. Effect of Au@Ag NRs on Inflammation and Dopamine Synthesis in Mouse Liver

The histological images revealed remarkable differences between the control and Au@Ag NRs treated mice. The structure of livers of mice in the control group was basically intact. Compared to the control, inflammatory infiltrates were detected around the vessel walls in the livers of Au@Ag NRs-treated mice (Figure 3A). The surrounding inflammatory infiltrates might mainly consist of monocytes and neutrophils as the previous publication reported [8]. Generally, these cells are considered as recruited from the circulating blood, demonstrating that treatment with Au@Ag NRs led to peripheral inflammation. In addition, there were some swelled hepatocytes and some hepatocytes with blurry cell membrane structures suffering cell necrosis, manifesting that Au@Ag NRs caused the hepatic injury.

As the above metabolic investigation showed, dopamine increased after Au@Ag NRs exposure, which might be related to the inflammation we observed. As is well known, tyrosine hydroxylase (TH) is the rate-limiting enzyme in the synthesis of dopamine [28]. The level of TH expression represents an ability to synthesize dopamine to some extent [29]. To reveal the effect of Au@Ag NRs on the synthesis of dopamine, immunohistochemical staining was used to investigate the TH level in mouse liver. The immunohistochemical results indicated the liver from Au@Ag NRs treated mice showed a higher TH level than that of control group (Figure 3B), suggesting that Au@Ag NRs indeed can induce the generation of dopamine in liver via increasing the level of TH. It was the first time to observe the production of dopamine in non-neural tissue based on the hints from metabolic results. Besides neurons, CD4^+^CD25^+^ regulatory T (Treg) cells can express TH and then assist in generating dopamine [30]. Herein, TH and dopamine might also result from the recruited and/or inherent Treg cells. Thus, Treg cells were obtained from female Balb/c mice to detect the effects of Au@Ag NRs on the secretion of dopamine in Treg cells. Indeed, Treg cells could secrete dopamine. However, treatment with Au@Ag NRs decreased the production of dopamine in Treg cells separated from the liver, as shown in Appendix A, which suggested that the increased production of dopamine in the liver was not directly generated by the Treg cells in response to Au@Ag NRs, but rather a product possibly synthesized in the liver for combating against Au@Ag NRs-induced inflammation or injury.

### 2.4. Cell Viability and Redox Balance Affected by Au@Ag NRs in Raw 264.7 Cells

As mentioned above, inflammatory cells in liver might be resulted from peripheral blood. Raw 264.7 cells, kinds of murine macrophages, were used to evaluate the cytotoxicity and related effects of Au@Ag NRs. Cells were treated with different concentrations of Au@Ag NRs (final concentration in the culture medium based on silver concentration) for 24 h. The cellular intake of Au@Ag NRs was characterized using two-photon luminescence (TPL) of Au@Ag NRs. Increasing exposure dose of Au@Ag NRs resulted in stronger TPL intensity in cells, suggesting more cellular uptake (Figure 4A). As shown in Figure 4B, the cell viability was maintained at a higher level than that of the control group at no more than 10 μg/mL, while the cell viability was significantly decreased once the silver concentration was up to 20 μg/mL. Thus, less than 20 μg/mL of silver concentration was optimal to further study.

Previous studies have proved that Ag NPs caused toxicity through the increasing reactive oxygen species (ROS). ROS are vital to cell signaling and cell growth. The exceeding ROS can disturb the intracellular redox balance and lead to oxidative stress. According to the Figure 4C, the total intracellular ROS level was measured, demonstrating that Au@Ag NRs induced the significant increase of ROS at the dosage of 10 μg/mL silver. GSH is an antioxidant and can scavenge the excess ROS. The change of reduced glutathione (GSH) level also reflects that the cellular redox homeostasis was affected (Figure 4D). As shown in Figure 4D, Au@Ag NRs caused a dose-depended change. When the silver concentration was 10 μg/mL, there was an obvious decrease in GSH level. Abnormally, when the silver concentration was as low as 1 μg/mL, the GSH level increased. Heme oxygenase 1 (HO-1), the rate-limiting enzyme in heme degradation, can also be employed to evaluate the level of oxidative stress. The Western blotting results showed Au@Ag NRs exposure triggered the upregulation of HO-1 when the silver concentration was 5 μg/mL or 10 μg/mL, and the HO-1 level increased in the dose-depended manner (Figure 4E). Due to the insufficient ROS, the silver concentration of as low as 1 μg/mL had no obvious effect on the HO-1 level.

### 2.5. Release of Proinflammatory Cytokines Enhanced by Au@Ag NRs in Raw264.7 Cells

The elevated ROS was a vital signal to activate inflammation. Lipopolysaccharide (LPS) was commonly used to induce the production of proinflammatory cytokines, including interleukin-1β (IL-1β) and interleukin-6 (IL-6) [31]. We investigate the inflammatory cytokines with LPS-activated macrophages. After treatment with Au@Ag NRs, IL-1β and IL-6 were measured using ELISA kits, respectively. As the results showed, Au@Ag NRs exposure caused the significantly rising production of IL-6 in a dose-dependent manner in LPS-activated macrophages (Figure 5A). Upon the silver concentration was up to 10 μg/mL, the secretion of IL-1β was also obviously increased compared with the control (Figure 5B).

### 2.6. Inflammation and Dopamine Synthesis in Lungs

Lungs are also one of the main target organs [32,33], after administration with Ag NPs. As shown in Table 1, seven days after multi-injections of Au@Ag NRs, silver still existed in the lungs. In the previous publication, after Au@Ag NRs were administrated via intravenous injection, both of the released Ag^+^ and NRs would also be deposited in the lungs and liver [22]. Animal exposure studies indicate that Ag NPs are able to cause the toxic effects in the lung. To determine whether Au@Ag NRs induced inflammation in the lungs, the expression level of IL-1β was visualized by immunohistochemistry of the lung sections. In the immunohistochemical data, a higher level of IL-1β was observed in the Au@Ag NRs-treated group (Figure 6A). Moreover, IL-1β was visualized around the vessels, which is similar to the location of inflammatory infiltrates in liver. Furthermore, to explore whether the increased synthesis of dopamine induced by Au@Ag NRs was the same with the liver, the level of tyrosine hydroxylase in the lungs was detected by immunohistochemistry. The results showed the lungs in the Au@Ag NRs treated mice expressed a little more tyrosine hydroxylase than the control (Figure 6B).

It hinted that the accumulation of Au@Ag NRs led to the increasing dopamine in the target organs, and dopamine increased with the development of inflammation. Dopamine is an agonist of dopaminergic receptors. It is reported that dopamine receptor D1 agonism promoted reversal of experimental lungs and liver fibrosis [34]. After treatment with Au@Ag NRs, the elevated production of endogenous dopamine might be related with tissue injury and repair. In the clinical, dopamine, as a kind of ‘‘first line” vasopressor to improve blood pressure and organ perfusion, has been used in patients with cirrhosis in the post-operative period [35].

## 3. Materials and Methods 

### 3.1. Chemicals Materials and Lab Animals

Acetonitrile, ammonium acetate, silver nitrate (AgNO_3_), sodium borohydride (NaBH_4_), tetrachloroauric acid (HAuCl_4_·3H_2_O), cetyltrimethyl ammonium bromide (CTAB), poly (sodium p-styrensulfonate) (PSS, MW 70,000), ascorbic acid (AA), 2′,7′-Dichlorofluorescin diacetate (DCFH-DA), and lipopolysaccharide (LPS) were purchased from Sigma-Aldrich (Munich, Germany). Cell Counting Kit-8 (CCK-8) was purchased from Dojindo Laboratories (Kumamoto, Japan). The resistivity of deionized water in this study was 18.2 MΩ.

All of the animal experiments involved herein were performed according to a protocol (ACUC-A02-2018-017, 20180315) that was approved by the Institutional Animal Care and Use committee (Institute of Basic Medical Sciences, Chinese Academy of Medical Sciences, and Peking Union Medical College, Beijing, China). 

### 3.2. Preparation and Characterization of Gold Nanorod Core/Silver Shell Nanostructures (Au@Ag NRs)

Gold nanorods (AuNRs) with LSPR at 720 nm were firstly synthesized according to previous publications. Then, 100 mL of as-prepared AuNRs was mixed with CTAB (10 mL, 0.1 M), ascorbic acid (10 mL, 0.1 M), and AgNO_3_ (1 mL, 0.01 M). Next, the mixed solution was stirred and kept in a 70 °C water bath for 8 h. To remove the extra CTAB and other ions, the nanoparticles were obtained by centrifuging (9000 rpm, 7 min). Afterwards, the nanoparticles were diluted in 100 mL water and co-incubated with 5 mL of 20 mg mL^−1^ polystyrene sulfonate (PSS) solution containing 60 mM NaCl overnight. Finally, the samples were centrifuged (12,000 rpm, 5 min), washed twice with pure water, and re-dispersed in pure water before the following experiments. The Ag/Au weight ratio in Au@Ag NR is 0.71 as determined by inductively coupled plasma mass spectrometer (ICP-MS) (NexION 300X, Perkin Elmer, Waltham, MA, USA).

To obtain the morphology and size of Au@Ag NRs, a transmission electron microscope (TEM, TecnaiG2 20 S-TWIN, Hillsboro, OR, USA) was used. According to TEM images, a software Image J (ImageJ 1.48, NIH, Bethesda, MD, USA) was used to measure the width and length of Au@Ag NRs. TEM element mapping of an individual Au@Ag NR was acquired from JEOL ARM200F (Tokyo, Japan). UV-Vis-NIR absorption spectra of Au@Ag NRs in water and 5% (m/m) glucose solution were measured using a Varian Cary 50 (Agilent, Palo Alto, CA, USA), respectively.

### 3.3. Intravenous Administration of Au@Ag NRs in Mice

Female Balb/c mice were injected with 50 μL of 0.5 mg mL^−1^ (silver concentration) Au@Ag NRs diluted in 5% isotonic glucose solution or 50 μL of 5% isotonic glucose solution alone (as negative control) on Day 1, 4 and 10. Each group consisted of 10 mice. Seven days after the last injection, all mice were sacrificed by cervical dislocation and organs (livers and lungs) were collected for further analysis. The dose of the nanostructures and duration of administration to the mice were referred to a previous study [8].

### 3.4. Metabolomics Analysis

Sample preparation: Liver tissue samples frozen at −80 °C were removed and thawed at room temperature. Then, 60 mg of liver tissue was mixed with 200 μL of water and was homogenized. In addition, 800 μL of methanol/acetonitrile mixture (1:1, *v*/*v*) was added. The mixture was fully mixed using vortex for 30 s and decomposed twice by ultrasonic crushing for 30 min at low temperature. After incubation for 1 h at −20 °C to precipitate protein, the mixture was centrifuged at 13,000 rpm for 15 min at 4 °C. The supernatant was collected, dried by lyophilization, and kept at −80 °C for further experiments.

LC-MS/MS analysis: The samples were separated using Agilent 1290 Infinity LC (Palo Alto, CA, USA) ultrahigh performance liquid chromatography (UHPLC) HILIC (hydrop interaction liquid chromatography) column (Waters). During the whole analysis process, the samples were placed in the automatic sampler at 4 °C to avoid the influence caused by the fluctuation of instrument detection signal. Samples were in a random order. Quality control samples were inserted into the sample queue for continuous analysis. After separation, the samples were analyzed with Triple TOF 5600 mass spectrometer (AB SCIEX, Concord, ON, Canada). During the process, electrospray ionization (ESI) positive ion and negative ion modes were used respectively.

Data analysis: The XCMS program (https://xcmsonline.scripps.edu/landing_page.php?pgcontent=mainPage) was used to correct the retention time of peak alignment and extract the structure of peak area metabolite. Accurate mass number matching (<25 ppM) and second-level spectrogram matching were used to retrieve the database of the laboratory (Shanghai Applied Protein Technology Co., Ltd., Shanghai, China). A software SIMCA-P 14.1 (Umetrics AB, Umea, Sweden) was furtherly used to perform univariate and multivariate statistical analysis.

### 3.5. Determination of Silver Content in Liver and Lung Tissues by ICP-MS

The tissue samples were freeze-dried and then weighed. Afterwards, the samples were predigested with HNO_3_ overnight and then heated with the addition of 30% H_2_O_2_ (*w*/*w*). After the sample solutions turned colorless, they were condensed to less than 1 mL. After that, the samples were diluted with 2% HNO_3_ (*w*/*w*) to a volume of 2 mL. Each group included four replicative samples, which were repeatedly measured three times.

### 3.6. Histopathological Evaluation and Immunohistochemical Test

After the collected organs were rinsed in phosphate buffer solution (PBS), the organs were fixed in 4% formaldehyde, embedded in paraffin, cut to sections, attached to glass slides, dewaxed, and hydrated. For histopathological evaluation, the sections of liver were then stained with haematoxylin and eosin (H&E staining) according to the standard procedures. For the immunohistochemical test, the sections of liver were blocked with 5% goat serum then probed with primary antibody against tyrosine hydroxylase. After washed, the sections were incubated with horseradish labeled secondary antibody. After washed, the sections were colored, re-dyed, dehydrated, transparent, and sealed. The sections of lung were blocked with 5% goat serum and then probed with a primary antibody against TH and IL-1β. After washed, the sections were incubated with a rhodamine labeled secondary antibody. The cell nucleuses were stained with 4’,6-diamidino-2-phenylindole (DAPI). The sections of liver were visualized with light microscopy (BX53, Olympus, Tokyo, Japan) with a CCD camera (DP72; Olympus). The sections of lung were visualized with fluorescence microscope (Olympus).

### 3.7. Treg Cells Isolation and Dopamine Assay

Spleens were removed from female Balb/c mice without any treatment and gently minced in complete dulbecco’s modified eagle medium (DMEM) containing 10% FBS. Then, the single cell suspensions were obtained by filtration. After removing red blood cells by lysis, CD4^+^CD25^+^ T cells were isolated and enriched by MagCellect Mouse CD4^+^ CD25^+^ Regulatory T Cell Isolation Kit (R&D Systems, Minneapolis, MN, USA) according to the detailed introduction. CD4^+^CD25^+^ T cells were cultured in completed DMEM medium and then exposed to Au@Ag NRs for 24 h. Each group was repeated three times. The supernatants were collected and the dopamine were determined by an Elisa kit (Elabscience, Wuhan, China).

### 3.8. Cell Culture and Cell Viability Assay

Raw264.7 cells were purchased from the Cell Resource Center of Chinese Academy of Medical Sciences (Beijing, China) and cultured in DMEM medium containing 10% (*v*/*v*) fetal bovine serum (FBS, GIBCO, Waltham, MA, USA) and streptomycin/penicillin (100 μg/mL or 100 U/mL, Hyclone, Logan, UT, USA) at 37 °C in a moisturized 5% CO_2_ incubator. A CCK-8 (Dojindo Laboratories in Kumamoto, Japan) assay was used to measured cell viability after Au@Ag NRs exposure. Cells were seeded in 96-well plates at a density of 20,000 cells per well and incubated overnight. Then, fresh medium containing different concentration of Au@Ag NRs was added and incubated with cells for 24 h. After incubation, the medium was replaced by fresh medium containing 10% (*v*/*v*) CCK-8. After 2 h incubation, a microplate reader (Tecan infnite M200, Männedorf, Switzerland) was used to read the absorbance at 450 nm with a reference at 600 nm. 

### 3.9. Two-Photon Luminescence (TPL) Imaging of Au@Ag NRs in Raw264.7 Cells

After incubation with Au@Ag NRs for 24 h, Raw264.7 cells were washed with PBS thrice. Then, TPL images of Au@Ag NRs within cells were obtained using a 60× oil lens on a confocal microscope system (FluoView1000, Olympus, Shinjuku, Japan) equipped with a femtosecond Ti: Sapphire laser (Mai Tai, Spectra-Physics, Santa Clara, CA, USA). The Au@Ag NRs were excited using a 690 nm NIR laser.

### 3.10. Intracellular Reactive Oxygen Species (ROS) Detection

After incubation with Au@Ag NRs for 24 h, RAW264.7 cells were collected and washed with PBS and then incubated with 10 μM DCFH-DA (Sigma) for 30 min at 37 °C before being subjected to a flow cytometer (Accuri C6, BD Biosciences, San Jose, CA, USA).

### 3.11. Enzyme-Linked Immunosorbent Assay (ELISA)

The supernatants of culture medium for RAW26.7 cells incubation with Au@Ag NRs together with LPS (Sigma) for 24 h were collected, and the concentration of IL-1β and IL-6 was determined by an ELISA kit (eBioscience, San Diego, CA, USA) according to the manufacture’s protocol.

### 3.12. Western Blotting

Thirty micrograms of protein of each sample was separated by sodium dodecyl sulfonate-polyacrylamide gel electrophoresis (SDS-PAGE) and then transferred to polyvinylidene difluoride membranes (Millipore, Billerica, CA, USA). Afterwards, blocking buffer, which was 5% non-fat milk diluted in TBS buffer containing 0.05% Tween-20 (TBST), was used to block the membranes at room temperature. After 1 h, the membranes were probed with primary antibodies (Cell Signaling Technologies, Boston, MA, USA) against β-actin and HO-1 overnight at 4 °C, respectively. After washing three times by TBST, the membranes were incubated with the corresponding secondary antibody in blocking solution for 1 h. After the membranes were washed five times with TBST, coloration with chemiluminescence was done and detected by an imaging system (Bio-Rad, Hercules, CA, USA).

## 4. Conclusions

In conclusion, treatment with Au@Ag NRs in vivo provoked inflammation, accompanied with alterations in metabolism (Figure 7). According to the metabolic profiling, we observed the changes in membrane and choline metabolism, energy metabolism, purine metabolism, redox metabolism, and dopamine synthesis. The change in dopamine synthesis was first found on non-neural tissues. All of them were related with inflammation. The histochemical results of liver and lung indicated the occurrence of the inflammation and injury. The inflammatory cells may come from peripheral blood. The toxicity and generation of ROS by exposure to Au@Ag NRs were evidenced in vitro by Raw 264.7 cells. The increase in IL-1β and IL-6 level showed the existence of inflammation after treating with Au@Ag NRs. Treg cells secreted dopamine in liver, but treatment with Au@Ag NRs did not upregulate dopamine in Treg cells. Hence, dopamine synthesis could be an active protection mechanism against oxidative stress-induced inflammation. It bridged a potent linkage among Ag NPs exposure, dopamine synthesis, and inflammation in vivo. It revealed the potent risks of Ag NPs for human health after long-term low-dose exposure. Furthermore, it hinted immune response containing the activation of inflammation as well as inhibition of inflammation to exogenous invaders, which provides more clues for nanoparticles to apply in biomedical field.

## Figures and Tables

**Figure 1 ijms-21-00384-f001:**
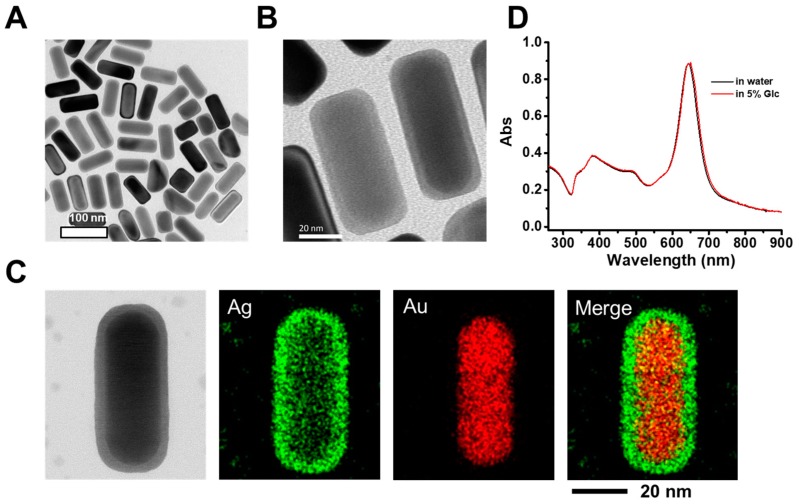
Characterizations of gold nanorods core/silver shell nanostructures (Au@Ag NRs). (**A**,**B**) Typical TEM images of Au@Ag NRs; (**C**) TEM image of an individual Au@Ag NR and its corresponding EDS (energy disperse spectroscopy) element maps of Au, Ag, and their overlay, respectively; (**D**) UV-vis-NIR (ultraviolet-visible-near infrared ray) extinction spectra of Au@Ag NRs dispersed in water and 5% glucose solution, respectively.

**Figure 2 ijms-21-00384-f002:**
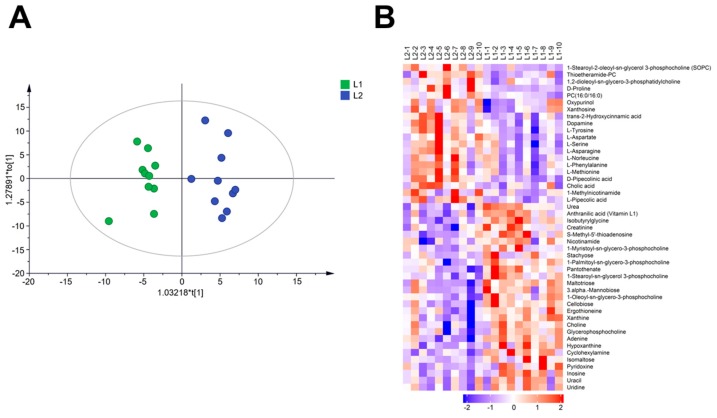
Effects of Au@Ag NRs on metabolic profiles of mouse liver. (**A**) Metabolic cluster analysis using an orthogonal partial least squares discrimination analysis (OPLS-DA) scores plot. The component t[1] is the predicted principal component, maximumly reflecting the inter-group differences, while intra-group variation is reflected in t0. L1-N is the control group and L2-N is the mice received multiple administration of Au@Ag NRs (N = 1–10); (**B**) heat maps of differential metabolites. All of the data were obtained under positive mode.

**Figure 3 ijms-21-00384-f003:**
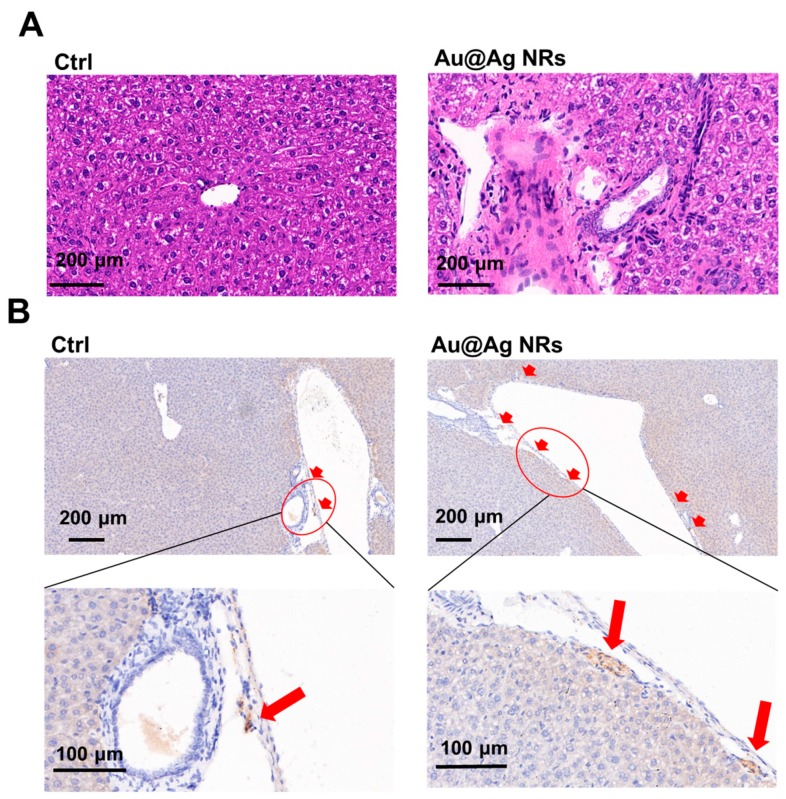
Influence of Au@Ag NRs on liver tissue. (**A**) histological images of liver from mice received multiple administration of Au@Ag NRs. The tissues were stained with hematoxylin and eosin; (**B**) the level of tyrosine hydroxylase (TH) expression in liver visualized by immunohistochemical staining, indicated by red arrows.

**Figure 4 ijms-21-00384-f004:**
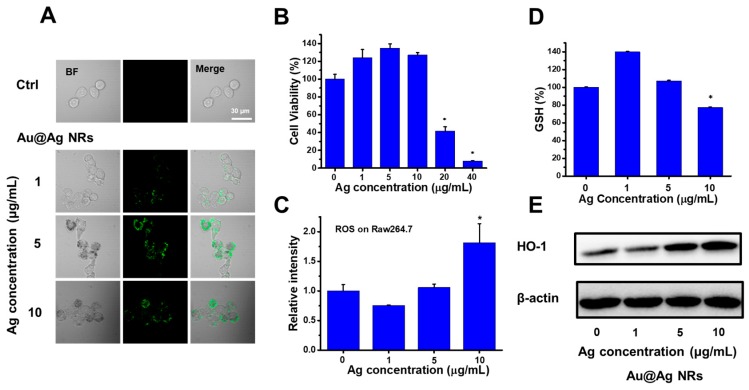
Effects of Au@Ag NRs on Raw264.7 cells. (**A**) in vitro two-photon luminescence (TPL) images upon different Au@Ag NR exposure concentrations; (**B**) cell viability of Raw264.7 cells affected by Au@Ag NRs; (**C**) folds of fluorescent intensity detected with 2′,7′-Dichlorofluorescin diacetate (DCFH-DA) by flow cytometry; (**D**) relative level of reduced glutathione (GSH) after the treatment with Au@Ag for 24 h; (**E**) Western blotting results of HO-1, after exposed to Au@Ag NRs for 24 h. The * represents significant difference from control group (*: *p* < 0.05). The data were presented as mean ± SD (*n* = 3).

**Figure 5 ijms-21-00384-f005:**
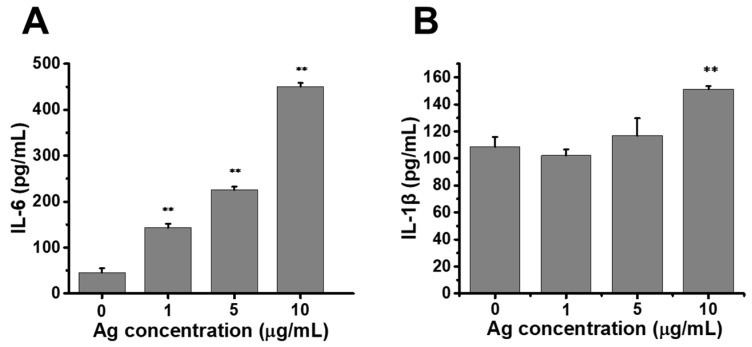
The secretion of inflammatory cytokines enhanced by Au@Ag NRs in Raw264.7 cells. (**A**) the secretion of IL-6, after 24 h exposure to Au@Ag NRs; (**B**) the IL-1β level in cultural supernatant. The * represents significant difference from control group (**: *p* < 0.01). The data were presented as mean ± SD (*n* = 3).

**Figure 6 ijms-21-00384-f006:**
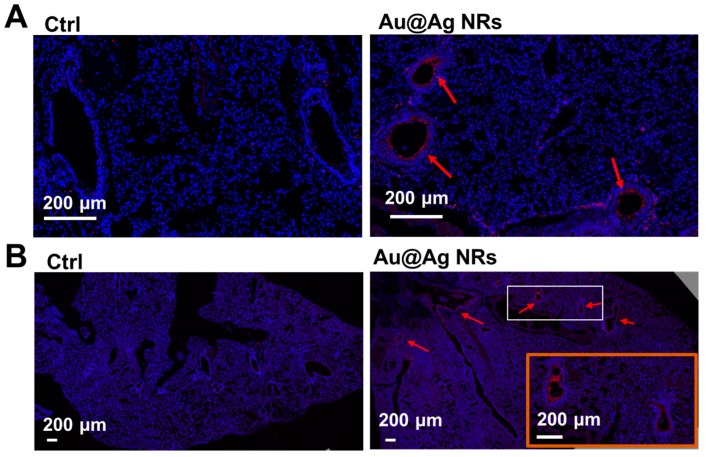
Influence of Au@Ag NRs on lung tissue. (**A**) IL-1β level in the lungs from mice received multiple administration of Au@Ag NRs. IL-1β was stained with red and indicated by red arrows, and the nucleus was stained with blue; (**B**) the level of tyrosine hydroxylase (TH) expression in the lungs visualized by immunofluorescence histochemistry. TH was stained with red and indicated by red arrows, and the nucleus was stained with blue. The inset in orange box is a magnification of the white box.

**Figure 7 ijms-21-00384-f007:**
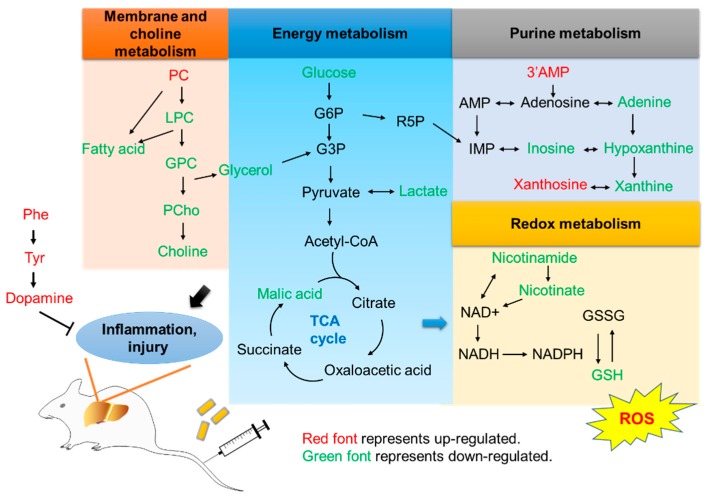
Summary of the main metabolic pathway alterations in the mouse liver upon exposure of Au@Ag NRs. Phosphatidylcholine (PC), lyso-phosphatidylcholine (LPC), glycerophosphocholine (GPC), phosphorylcholine (PCho), phenylalanine (Phe), tyrosine (Tyr)**.**

**Table 1 ijms-21-00384-t001:** The content of silver in the organs of mice after receiving multi-dose administration of gold nanorods core/silver shell nanostructures (Au@Ag NRs) analyzed by inductively coupled plasma mass spectrometer (ICP-MS).

Organs	Ag Content (μg/g)
liver	49.31 ± 11.65
lung	9.61 ± 3.08

The data were presented as mean ± standard deviation (SD) (four mice in each group). Tissue samples of liver and lung were collected on day 7 after the last injection. Silver amount in the control group was not detected.

**Table 2 ijms-21-00384-t002:** Variant metabolites from different metabolic functions and pathways.

Mode	Metabolites	Trend	VIP	*p*-Value	Pathway
negative	Alpha-D-Glucose	down	18.5136	0.014065	Energy metabolism
negative	L-Malic acid	down	3.14375	0.002261	
negative	DL-lactate	down	3.10708	0.006937	
negative	Glycerol	down	1.83996	0.073435	
positive	PC(16:0/16:0)	up	4.06947	0.019537	Membrane metabolism and choline metabolism
positive	Thioetheramide-PC	up	5.29022	0.012911	
positive	1-Stearoyl-sn-glycerol-3-phosphocholine	down	7.59854	0.002209	
positive	1,2-dioleoyl-sn-glycero-3-phosphatidylcholine	up	2.33406	0.067463	
positive	1-Myristoyl-sn-glycero-3-phosphocholine	down	1.90661	0.07697	
positive	1-Oleoyl-sn-glycero-3-phosphocholine	down	7.27013	0.005697	
positive	1-Palmitoyl-sn-glycero-3-phosphocholine	down	6.09609	0.011949	
positive	1-Stearoyl-2-oleoyl-sn-glycerol 3-phosphocholine (SOPC)	up	3.54098	0.003904	
positive	Glycerophosphocholine	down	6.89478	0.022855	
positive	Phosphorylcholine	down	1.79248	0.003768	
positive	Choline	down	1.0697	0.020519	
positive	L-Phenylalanine	up	5.14322	0.078525	Dopamine synthesis
positive	L-Tyrosine	up	2.80009	0.006615	
positive	Dopamine	up	1.19994	0.003964	
positive	Cholic acid	up	5.21803	0.049867	cholic acid synthesis
negative	Chenodeoxycholate	up	2.19638	0.097443	
positive	Hypoxanthine	down	2.56096	0.005989	Purine metabolism
positive	Xanthine	down	3.20653	0.000404	
positive	Adenine	down	2.30405	0.000537	
positive	Inosine	down	4.63986	0.003509	
negative	Adenosine 3′-monophosphate	up	1.01864	0.078576	
positive	Nicotinamide	down	6.21993	0.0432	Redox metabolism
negative	Nicotinate	down	1.06332	0.003276	

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
