# Peer review of "In Vivo Metabolic Response upon Exposure to Gold Nanorod Core/Silver Shell Nanostructures: Modulation of Inflammation and Upregulation of Dopamine"

_ijms, 2020, doi:10.3390/ijms21020384_

Round 1

Reviewer 1 Report

The article "In vivo metabolic response upon exposure to gold nanorod core/silver shell nanostructures: modulation of inflammation and upregulation of dopamine" by Li H. et al. is devoted to an urgent study of the toxic effects of intravenous injection of Au@AgNRs on murine Raw264.7 cells and on liver and lung cells of mice. The authors showed that Au@AgNRs cause oxidative stress, which induces inflammation. They used adequate methods (analysis of metabolism, measurement of the main indicators of oxidative stress and inflammation) and convincingly showed that the concentration of intermediate products of energy metabolism decreased, the level of pro-inflammatory cytokines and the content of dopamine in the liver and lungs increased. All sections of the article are described clearly.

Major point

The mice were treated with Au@AgNRs on the next scheme. (Lanes 276 - 279) Female Balb / c mice were injected with 50 μL of 0.5 mg mL-1 (silver concentration) Au @ Ag NRs diluted in 5% isotonic glucose solution or 50 μL of 5% isotonic glucose solution alone (as negative control) on Day 1, 4 and 10. Each group consisted of 10 mice. Seven days after the last injection, all mice were sacrificed by cervical dislocation and organs (livers and lungs) were collected for further analysis. According to this scheme, a week passes between 4 and 10 days. Since there is a clearance of silver, there can be no difference between a single administration and 3-fold injections. Authors should check the silver content in the tissues, in which were analyzed during the experiment, and these data must be added to article.

Minor points

The authors demonstrate the dependence of the survival of Raw264.7 cells on Au @ Ag) NRs concentration. However, it is not shown whether silver enters the cells. In addition, the authors do not provide data on the intake of silver in the liver and lungs. The absence of these data makes it difficult to understand the action of nanomaterial. Lanes 257 - 259: Protocol No. and date must be added. Line 108 – Phospboryicholine (?) AgNPs are recorded non-uniformly, for example, Fig. 1 line 63: AgNPs; Lane 70: Ag NPs. To fix.

Author Response

Major points

The mice were treated with Au@AgNRs on the next scheme. (Lanes 276 - 279) Female Balb / c mice were injected with 50 μL of 0.5 mg mL-1 (silver concentration) Au @ Ag NRs diluted in 5% isotonic glucose solution or 50 μL of 5% isotonic glucose solution alone (as negative control) on Day 1, 4 and 10. Each group consisted of 10 mice. Seven days after the last injection, all mice were sacrificed by cervical dislocation and organs (livers and lungs) were collected for further analysis. According to this scheme, a week passes between 4 and 10 days. Since there is a clearance of silver, there can be no difference between a single administration and 3-fold injections. Authors should check the silver content in the tissues, in which were analyzed during the experiment, and these data must be added to article.

Response: Thanks for the referee’s suggestion. Contents of silver in liver and lung tissues of the mice have been measured using ICP-MS and added in the revised manuscript as Table 1.

Table 1. The content of silver in the organs of mice after receiving multi-dose administration of Au@Ag NRs analyzed by ICP-MS.

Organs

Ag content (μg/g)

liver

49.31 ± 11.65

lung

9.61 ± 3.08

The data were presented as mean ± SD (four mice in each group). Tissue samples of liver and lung were collected on the day 7 after the last injection. The content of silver in the control group were not detected.

Minor points

The authors demonstrate the dependence of the survival of Raw264.7 cells on Au @ Ag) NRs concentration. However, it is not shown whether silver enters the cells. In addition, the authors do not provide data on the intake of silver in the liver and lungs. The absence of these data makes it difficult to understand the action of nanomaterial. Lanes 257 - 259: Protocol No. and date must be added. Line 108 – Phospboryicholine (?) AgNPs are recorded non-uniformly, for example, Fig. 1 line 63: AgNPs; Lane 70: Ag NPs. To fix.

Response:

1) Taking the advantage of strong two-photon luminescence (TPL) of Au@AgNRs, we used in vitro TPL images of NRs to demonstrate the uptake of the NRs by Raw264.7 cells, which have been added as Figure 4A. The results of silver content in tissues of liver and lung have been displayed in Table 1. These two experiments verified the uptake of silver in the liver and lungs.

2) Protocol No. and date have been added (Line 288-289).

3) The word “Phospboryicholine” has been corrected to “Phosphorylcholine” (Line 131).

4) Ag NPs have been recorded uniformly (marked red in the text).

Reviewer 2 Report

In this paper, the researchers investigated the effects of gold nanorod core/silver shell nanostructures (Au@Ag NRs) on the liver and macrophage through in vitro and in vivo experiments. Here are the comments made by this Reviewer.

Major issues:

The authors need to include a scientific rationale for using nanorods over nanoparticles, preferably in the introduction section. Figure 1: Please include the size distribution of the synthesized NRs. Why did the author choose such sizes and aspect ratios for conducting this study? This is an important issue since the size-dependent effects of nanoparticles/nanorods have been intensively studied in a number of previous reports. Please explain why the authors used a 5% glucose solution. The scientific importance of a narrow LSPR band in Figure 1C also should be properly explained. How was the nanostructure sterilized before it was applied to cells and animals? Please explain. How did the authors determine the dose of the NP and duration of administration to the mice? Please address this issue. The authors should conduct statistical analysis for Figure 4A and Figure 5A. The method and specific details (e.g. number of samples used for replication, types) also need to be clearly explained.

Minor issues:

The scale bar shown in Figure 1A is not very clear. Please correct this. While Figure 2 and Table S1 were obtained with the positive mode, why was the negative mode used in Figure S1? Please explain why.

Author Response

Major points

The authors need to include a scientific rationale for using nanorods over nanoparticles, preferably in the introduction section. Figure 1: Please include the size distribution of the synthesized NRs. Why did the author choose such sizes and aspect ratios for conducting this study? This is an important issue since the size-dependent effects of nanoparticles/nanorods have been intensively studied in a number of previous reports. Please explain why the authors used a 5% glucose solution. The scientific importance of a narrow LSPR band in Figure 1C also should be properly explained. How was the nanostructure sterilized before it was applied to cells and animals? Please explain. How did the authors determine the dose of the NP and duration of administration to the mice? Please address this issue. The authors should conduct statistical analysis for Figure 4A and Figure 5A. The method and specific details (e.g. number of samples used for replication, types) also need to be clearly explained.

Response:

1) The scientific rationale for using nanorods over nanoparticles has been added in the introduction section. The reasons are as follows: a) Compared to spherical NPs, NRs have aspect ratio-dependent local plasmonic features, which can be extended to NIR spectral region and thus are beneficial for biological applications. b) Compared to spherical shape, the unique rod-shape makes it more easily identify in vitro.

2) The mean aspect ratio of NRs and its distribution has been added (Figure S1).

3) The reasons for choosing such sizes and aspect ratios in this study have been added in the text. Au@Ag NRs with sizes similar to a previous study of bio-distribution and bio-availability of Au@Ag NRs in rat tissues were used as they were shown to accumulate in organs such as liver and lung by intravenous injection [23].

4) In general, stroke-physiological saline solution and 5% glucose solution are treated as solvents for animal experiments. In view that the chloride ions in stroke-physiological saline solution may affect the silver dissolution, 5% glucose solution was chosen herein as a proper solvent. As shown in Figure 1D, nanorods dispersed in 5% glucose solution showed a good dispersion stability.

5) The reason of a narrow LSPR band in Figure 1D has been properly explained. Owing to “plasmonic-focusing” effect [22], Au@Ag NRs showed a narrower LSPR band compared to inside Au NR cores, which is beneficial for many plasmonic applications such as sensing and multiplex labeling.

6) The conventional sterilization (autoclaving, ultraviolet irradiation, and filtration) is not suitable for our nanostructures. Therefore, we have tried to avoid contamination during the process of nanostructures preparation. The nanostructures were prepared in a clean room and the deionized water was sterilized before use. The sterilized 5% glucose solution was used to dilute the nanostructures and the procedure has been completed in an ultra-clean bench.

7) The selection for the dose of the NP and duration of administration to the mice have been explained in the section of methods. The dose of the nanostructures and duration of administration to the mice were determined according to a previous reference [8].

8) The statistical analysis has been added in Figure 4B and Figure 5A.

9) The method and specific details (e.g. number of samples used for replication, types) have been clearly explained (marked in red in the text).

Minor issues:

The scale bar shown in Figure 1A is not very clear. Please correct this. While Figure 2 and Table S1 were obtained with the positive mode, why was the negative mode used in Figure S1? Please explain why

Response:

1) The scale bar shown in Figure 1A has been re-drawn.

2) A few metabolites were not detected under positive mode but detected under negative mode. Therefore, the results with the negative mode were shown in Figure S1 to supplement the alterations of metabolites caused by nanostructures.

Reviewer 3 Report

the manuscript "In vivo metabolic response upon exposure to gold
nanorods core/silver shell nanostructures: modulation
of inflammation and upregulation of dopamine" is an interesting study, but there are some concerns:

-NPs were characterized with TEM and UV-VIS. The core/shell structure costituing by Au and AG. In this way it is important to analyse qualitatively the presence of the two metals.

-Why the authors choose the Female Balb/c mice? they should explain in the text.

-Dose selection for in vitro/in vivo studies: The rationale for dose selection is -absent

-The introduction should be integrate with some references to focalize the effects in cells:

-Int J Mol Sci. 2019 Jan; 20(2): 449. doi: 10.3390/ijms20020449

- J Nanopart Res (2018) 20: 273. https://doi.org/10.1007/s11051-018-4383-3

-toxicology reports, vol 2 2015, Pages 574-579  https://doi.org/10.1016/j.toxrep.2015.03.004,

-https://doi.org/10.2147/IJN.S174515

Author Response

Point 1: NPs were characterized with TEM and UV-VIS. The core/shell structure costituing by Au and AG. In this way it is important to analyse qualitatively the presence of the two metals.

Response 1: EDS element mapping of Au and Ag in a single NR has been added in Figure 1C to qualitatively show the presence of the two metals.

Point 2: Why the authors choose the Female Balb/c mice? they should explain in the text.

Response 2: Our study has no specific requirements for mice genotypes or phenotypes. Female Balb/c mice with small individual differences have no mutual invasion and are easy to herd.

Point 3: Dose selection for in vitro/in vivo studies: The rationale for dose selection is -absent.

Response 3: Dose selection for in vitro (Line 197-200)/in vivo (Line 316-317) studies has been explained in the text.

Point 4: The introduction should be integrate with some references to focalize the effects in cells:

-Int J Mol Sci. 2019 Jan; 20(2): 449. doi: 10.3390/ijms20020449

- J Nanopart Res (2018) 20: 273. https://doi.org/10.1007/s11051-018-4383-3

-toxicology reports, vol 2 2015, Pages 574-579  https://doi.org/10.1016/j.toxrep.2015.03.004,

-https://doi.org/10.2147/IJN.S174515

Response 4: The recommended references have been added in the introduction (Reference 2,6,7,9).

Round 2

Reviewer 1 Report

The authors took into account all the comments and made changes to the manuscript. In its present form, the article may be accepted for publication.

Author Response

Point: The authors took into account all the comments and made changes to the manuscript. In its present form, the article may be accepted for publication.

Response: Thank you for your comments.

Reviewer 2 Report

The authors responded to all comments made by this Reviewer and revised the manuscript accordingly. Therefore, the manuscript is now acceptable for publication in the journal.

Author Response

Thank you for your comments.

Reviewer 3 Report

I recommend the publication of the manuscript after typos errors correction.

Author Response

Thank you for your comments.  The typos errors have been revised.

In line 24, the word “its” has been revised to “the”.

The order of reference 23 and reference 22 has been adjusted.

In line 244, “lung” has been revised to “liver”.

In line 44-45, 265 and 276, “in vivo” has been revised to “in vivo” in italic type.

In line 75, “in vitro” has been revised to “in vitro” in italic type.